# Influence of Climate Changes on the State of Water Resources in Poland and Their Usage

**Katarzyna Kubiak-Wójcicka** [1],* **and Sylwia Machula** [2]

[1] Department of Hydrology and Water Management, Faculty of Earth Sciences and Spatial Management, Nicolaus Copernicus University, Lwowska 1, 87-100 Toruń, Poland
[2] Department of Commodity Science, Quality Assessment, Process Engineering and Human Nutrition, West Pomeranian University of Technology in Szczecin, K. Królewicza 4, 71-550 Szczecin, Poland; Sylwia.Machula@zut.edu.pl
* Correspondence: kubiak@umk.pl

**Abstract:** The study aims to estimate the amount of available renewable water resources in Poland in the years 1999–2018 and the extent of their use by various sectors of the national economy at the national and regional levels. In the study period, the selected meteorological elements were found to have changed, resulting in a decrease in the flows of the two largest rivers in Poland: the Vistula and the Oder. The outflow of the Vistula and Odra basins determines the size of Poland's water resources. Poland is classified as a country of low water resources, as evidenced by the per capita amount of surface water, which in the years 1999–2018 was 1566 m$^3$/capita. Water consumption to meet the needs of the economy and the population was stable, and averaged 283 m$^3$/capita in this period. The analysis of water consumption by region showed that the areas with the lowest annual precipitation consume significant amounts of water for economic purposes, which may limit or destabilise socio-economic development in the region in future. Based on the difference between the amount of precipitation and water losses in the form of evaporation and water abstraction for economic purposes, maps were drawn up showing the deficit of surface water in a dry year. During periods of surface water scarcity, groundwater uptake increases. An area particularly exposed to water scarcity is central Poland.

**Keywords:** climate change; water resources; water consumption; Poland

## 1. Introduction

Freshwater resources are widely recognised as essential for mankind and the ecosystem [1–4]. The sustainable use of water resources has been increasingly challenging in recent years due to increasing water demand and climate change. According to the IPCC [5], significant global warming and changes in extreme weather events confirm global climate change. As a consequence of global warming, changes in the hydrological cycle will take place [6]. The impact of these phenomena will have different effects in different regions of the world. Climate change is predicted to increase water shortages. Water scarcity can generally be described as the scarcity of renewable freshwater availability in relation to demand [7]. Population distribution, climate and hydrological conditions vary considerably worldwide [8], and there is often a gap between water demand and supply [9–11]. Both developed and developing economies face an imbalance between water supply and demand [12–14]. Freshwater shortages directly affect food safety, access to safe drinking water, public health and hygiene, and environmental well-being [15]. Water scarcity can also delay economic development and foster civil conflict [4]. In recent years, various studies have indicated excessive consumption of surface water and groundwater resources [16–18]. Water management deals not only

with the problem of its lack or periodic shortages and the quality enabling or limiting its use, but also with the maintenance of aquatic and other related ecosystems [19].

According to the European Environment Agency [20], the balance between water demand and availability has reached critical levels in many areas of Europe as a result of over-exploitation and prolonged periods of low precipitation or drought. The most severe impacts are expected in the southern and south-eastern regions of Europe, which are already suffering from water scarcity. These areas will face reduced water availability as more frequent and more intense droughts occur. While water availability will generally increase in northern regions, it may decrease during summer periods and lead to periods of drought [21].

In light of these considerations, Poland is one of the countries of Europe with the smallest water resources per capita. Water shortage may have far-reaching consequences. In the last few years, Poland has experienced meteorological droughts, which have contributed to water shortages across most of its territory [22]. Reduced flow in rivers and streams may increase the concentration of harmful pollutants [23–25].

Taking into account the above, the main objective of this study was to estimate the amount of available renewable water resources in Poland in the period 1999–2018 and the degree of their use by various branches of the national economy in national and regional terms. The size of these resources is strictly dependent on meteorological conditions, which have been changing in recent years. The aim of this study is to provoke a discussion on sustainable management of water resources during periods of drought and the possibility of adapting to future climate changes.

## 2. Materials and Methods

### 2.1. Research Area

The total area of the country is 312,722 km$^2$ [26]. The population living in Poland at the end of December 2018 was 38.411 million. In the years 1999–2018, the population in Poland ranged from 38.116 million (2007) to 38.538 million (2011). The largest numbers of people live in Mazowieckie and Śląskie Voivodeships (administrative units), and the smallest in Opolskie Voivodeship. The average population density in Poland is 123 persons/km$^2$ and remained stable in the analysed period. Relative to other countries of the European Union (117 persons/km$^2$) this value is average. The highest average population density in Poland at the end of 2018 was recorded in Śląskie (368 persons/km$^2$), Małopolskie (224 persons/km$^2$) and Mazowieckie (152 persons/km$^2$). The voivodeships with the lowest population density were Podlaskie and Warmińsko-Mazurskie, where the population density was 59 persons/km$^2$. The highest population density is recorded in cities. In the ten most densely populated cities there are over 3000 persons/km$^2$. Those cities are: Warsaw (population 1,777,972), Kraków (771,069), Łódź (685,285), Wrocław (640,648), Poznań, Gdańsk, Szczecin, Bydgoszcz, Lublin and Katowice—as of 31 December 2018 [26].

Poland is a lowland country, with almost 50% of the country's area at elevations of between 100 and 200 m above sea level. Mountains (above 500 m) constitute only 3% of the country's area.

Poland has a temperate climate with a transitional character between the maritime and continental. In the west of the country, there is a significant predominance of oceanic influences, i.e., smaller annual temperature amplitudes, early spring and summer, and short winter. To the east, there is a predominance of continental influences characterised by higher annual temperature amplitudes, short hot summers and long, cold winters. In the north of Poland, there is a predominance of maritime influences, with short, not very hot summers and long, warm winters. In the south, however, there is a predominance of highland and mountain influences, with heavy precipitation and a wide variety of local climatic conditions. The maximum precipitation falls in the summer months (June–August), while the minimum precipitation occurs from January to March [27].

Inland surface waters are directly supplied by precipitation and indirectly by meltwaters. The most important elements of the water network in Poland are: rivers, lakes, ponds, underground waters,

man-made reservoirs and canals. Poland is located almost entirely in the Baltic Sea basin (99.7% of the country's area). Among others, it includes the basins of the largest rivers: the Vistula and the Oder. The remaining 0.3% of the country's territory is covered by basins of rivers entering the Black Sea (0.2%) and North Sea (0.1%) catchment systems. Most of the rivers in Poland flow north-westwards, according to the slope of the country's surface.

Nearly 88% of Poland's total area lies in the basins of the two largest Polish rivers: the Vistula and the Oder. The Vistula river basin (excluding the delta) covers an area of 194.0 thousand km$^2$, of which 168.9 km$^2$ is on Polish territory. The length of the Vistula is 1022 km. The Oder river basin covers an area of 119.1 thousand km$^2$, of which 106 thousand km$^2$ is on the territory of Poland. The length of this river is 840 km (including 726 km in Poland).

### 2.2. Data and Methods

In order to identify the impact of meteorological conditions on changes in water resources and the degree of their use in the economy, hydrometeorological and statistical data were analysed. The study included annual sums of precipitation and average annual air temperatures in the years 1999–2018 for 12 selected meteorological stations. The size of water resources was presented based on hydrological data from two hydrological stations. The stations are located in estuarial sections of the largest Polish rivers, i.e., the Vistula River—the station in Tczew, and the Oder River—the Gozdowice station. The data included annual and monthly average values from the period 1999–2018. The data come from the Institute of Meteorology and Water Management—National Research Institute.

To characterise Poland's water resources and the way they are used, the volume of water intake by three sectors of the national economy (agriculture, industry, and operation of the water supply network), total water consumption per capita and sources of water intake are presented. These values were obtained from the Local Data Bank of the Central Statistical Office (https://bdl.stat.gov.pl/BDL/start) and from the 'Environment Protection' annuals covering the period 1999–2019. The data were presented in general for Poland and in the regional perspective, taking into account the division of the country into administrative units (voivodeships). A detailed analysis was carried out in three selected years, i.e., 2010, 2015 and 2018. The year 2010 was characterised by the highest amount of precipitation in Poland (wet year), while the years 2015 and 2018 were the years with the lowest amount of precipitation (dry years) in the period 1999–2018. The universal WSI (Water Stress Index) indicator, which determines the amount of available resources and the Water Exploitation Index (WEI), which determines the ratio of the amount of water consumed to the total water resources, were also used in the work.

To show the regional diversification of water resources in Poland, a water deficit map was presented for a dry year (2015). The water deficit map for individual voivodeships takes into account the renewable resources of surface waters, calculated as the annual sum of precipitation, minus average annual evaporation and water abstracted for economic needs. The water resources within a given voivodeship were quantified based on the distribution of precipitation in 2015. This was done by calculating the area between the lines defining the annual sums of precipitation was calculated. Then, the cut-off values or precipitation intervals were averaged and multiplied by the area occupied, and then summed to obtain the amount of a given voivodeship's own water resources (in km$^3$). The average annual sum of evaporation in Poland ranges from 380 to 550 mm [28]. The adopted evaporation values are consistent with the results of other authors [29,30]. After calculating the amount of evaporation (minimum 380 mm or maximum 550 mm) from the area of individual provinces, the amount of evaporation in km$^3$ was obtained. The water deficit of individual voivodeships was quantified by taking the amount of precipitation and subtracting the amount of evaporation and the amount of water extraction for the area of a given voivodeship.

The research period 1999–2018 was selected due to the availability of statistical data related to water intake for economic needs in accordance with the current administrative division of the country from 1 January 1999.

## 3. Results

### 3.1. Precipitation in Poland and Its Variability

On the basis of annual sums of precipitation, the spatial distribution of precipitation in Poland in the years 1999–2018 was presented (Figure 1). The spatial diversity of annual sums of precipitation determining the size of water resources in Poland is very high and ranged from about 500 mm to 1170 mm. The lowest precipitation in the analysed period was recorded in Wielkopolska and central Poland, where annual sums of precipitation were below 550 mm. In the remaining area of the lowlands, annual sums of precipitation ranged from 550 to 600 mm. Higher sums of precipitation were recorded in the northern and southern parts of Poland. In the north-eastern part of the country, the annual sums of precipitation ranged from 600 to 650 mm, while in the north-western part, the annual sums of precipitation were higher than in the Mazurian Lake District and ranged from 600 to 750 mm. In upland areas, total precipitation was about 600 mm, while in mountain and foothill areas it was over 800 mm, and in high parts of the mountains it reached over 1100 mm.

For detailed analysis of meteorological conditions, 12 meteorological stations were selected in different regions of Poland (Table 1, Figure 1). These stations represent regions with different physical-geographical, natural and economic conditions. The highest annual amounts of precipitation in Poland in the period 1999–2018 were recorded in 2010 at most of the meteorological stations, while the lowest annual amounts of precipitation were recorded in 2015. The intense precipitation that took place in May and June 2010 caused the largest flooding in the Vistula basin in recent years [31]. In turn, low amounts of precipitation in 2015 contributed to a widespread drought, which took place not only in Poland but almost all of Europe [32]. Low amounts of precipitation also occurred in 2018.

**Table 1.** Average annual precipitation and air temperature at selected meteorological stations in Poland in period 1999–2018 (based on Institute of Meteorology and Water Management—National Research Institute data).

| Meteorological Stations | Altitude (m a.s.l.) | Average Annual Precipitation (mm) | | | | Average Annual Air Temperature (°C) | | | |
|---|---|---|---|---|---|---|---|---|---|
| | | 1999–2018 | 2010 | 2015 | 2018 | 1999–2018 | 2010 | 2015 | 2018 |
| Chojnice | 164 | 629 | 800 | 433 | 520 | 8.2 | 6.6 | 9.0 | 9.1 |
| Szczecin | 1 | 576 | 716 | 438 | 401 | 9.6 | 8.0 | 10.2 | 10.3 |
| Białystok | 148 | 614 | 851 | 526 | 536 | 7.8 | 6.8 | 8.6 | 8.7 |
| Toruń | 69 | 565 | 832 | 379 | 411 | 9.1 | 7.4 | 9.9 | 10.2 |
| Poznań | 87 | 547 | 715 | 438 | 373 | 9.5 | 7.7 | 10.4 | 10.7 |
| Warsaw | 106 | 557 | 798 | 404 | 433 | 9.2 | 8.0 | 10.3 | 10.4 |
| Łódź | 187 | 598 | 751 | 396 | 520 | 8.9 | 7.5 | 9.9 | 9.8 |
| Lublin | 238 | 611 | 751 | 532 | 479 | 8.4 | 7.5 | 9.4 | 9.3 |
| Wrocław | 120 | 535 | 692 | 388 | 398 | 9.9 | 8.2 | 11.1 | 11.2 |
| Kielce | 260 | 636 | 744 | 557 | 487 | 8.4 | 7.3 | 9.3 | 9.4 |
| Kraków | 237 | 681 | 1021 | 551 | 569 | 9.0 | 7.5 | 10.0 | 10.0 |
| Bielsko-Biała | 398 | 1005 | 1478 | 768 | 805 | 9.2 | 7.9 | 10.2 | 10.0 |

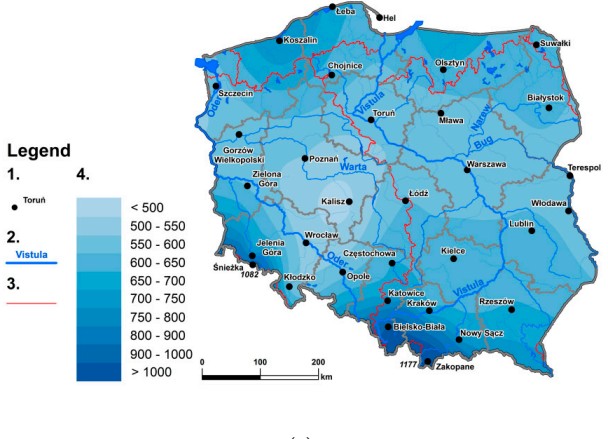

(a)

**Figure 1.** *Cont.*

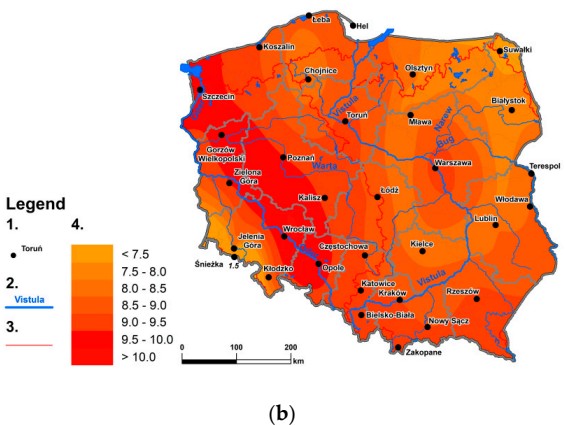

(**b**)

**Figure 1.** Spatial distribution of average amounts of precipitation (**a**) and air temperature (**b**) in Poland in the period 1999–2018 (based on Institute of Meteorology and Water Management—National Research Institute data). Legend (**a**): 1—meteorological stations, 2—rivers, 3—river basin border, 4—average annual precipitation in mm; (**b**) 1—meteorological stations, 2—rivers, 3—river basin border, 4—average annual air temperature in °C.

In addition to annual precipitation totals, the average air temperature is an important factor influencing the size of water resources. The average annual air temperature over the years 1999–2018 ranged from 7.8 °C in north-eastern Poland (Białystok) to 9.9 °C in western Poland (Wrocław). In meteorological stations located in the centre of Poland, the average annual air temperature was between 9.0 °C and 9.5 °C. The year 2010 was characterised by significantly lower annual air temperatures (from 6.6 °C in the north of the country in Chojnice to 8.2 °C in Wrocław). Markedly higher annual air temperature values were recorded in 2015 and 2018, as well as a greater difference between the analysed meteorological stations. It was warmest then in Wrocław, and coldest in Białystok.

The distribution of annual and monthly precipitation and average air temperature in the years 1999–2018 was presented for four selected meteorological stations (Figures 2 and 3). The selection of these stations was dictated by their location in a similar geographical longitude and at the same time reflected the conditions characteristic for regions with different physical and geophysical characteristics. In the analysed period, increasing trends in average air temperatures are visible. The growth rate is different: from 0.25 °C to 0.40 °C per decade, which indicates significant warming in this 20-year period. Symptoms indicating systematic and long-term warming in the years 1951–2015 in Poland are confirmed by research conducted by the authors [33]. In the case of precipitation, most of the meteorological stations recorded decreasing trends or no trend in the course of annual precipitation sums. The study [34] for a longer research period (1951–2013) showed that annual sums of precipitation increased slightly in Poland. However, these changes were not statistically significant in the whole area. Low precipitation was often accompanied by high air temperature, which resulted in an increase in evaporation and, as a consequence, meteorological drought [35]. Long-term meteorological drought adversely affected the surface water supply, which led to hydrological drought.

The distribution of precipitation and air temperature in annual terms is of a great importance, as it shows significant seasonal variability, especially during the summer. From May to September, there is relatively high precipitation in Poland, but, as a result of high air temperature, there is a large evaporation and consequently a decrease in surface water resources. This period coincides with the growing season, when there is the greatest demand for water in agriculture.

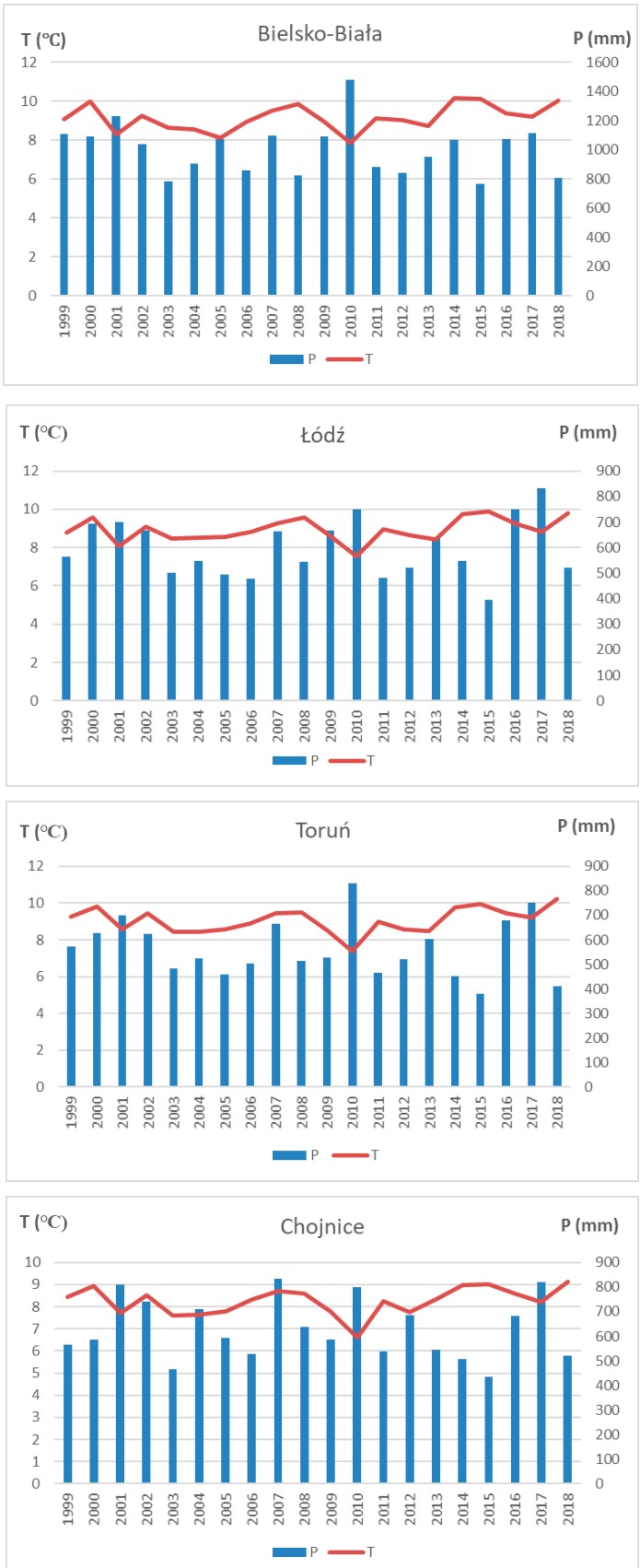

**Figure 2.** Average annual air temperatures (T) and precipitation (P) at selected meteorological stations in Poland in the period 1999–2018 (based on Institute of Meteorology and Water Management—National Research Institute data).

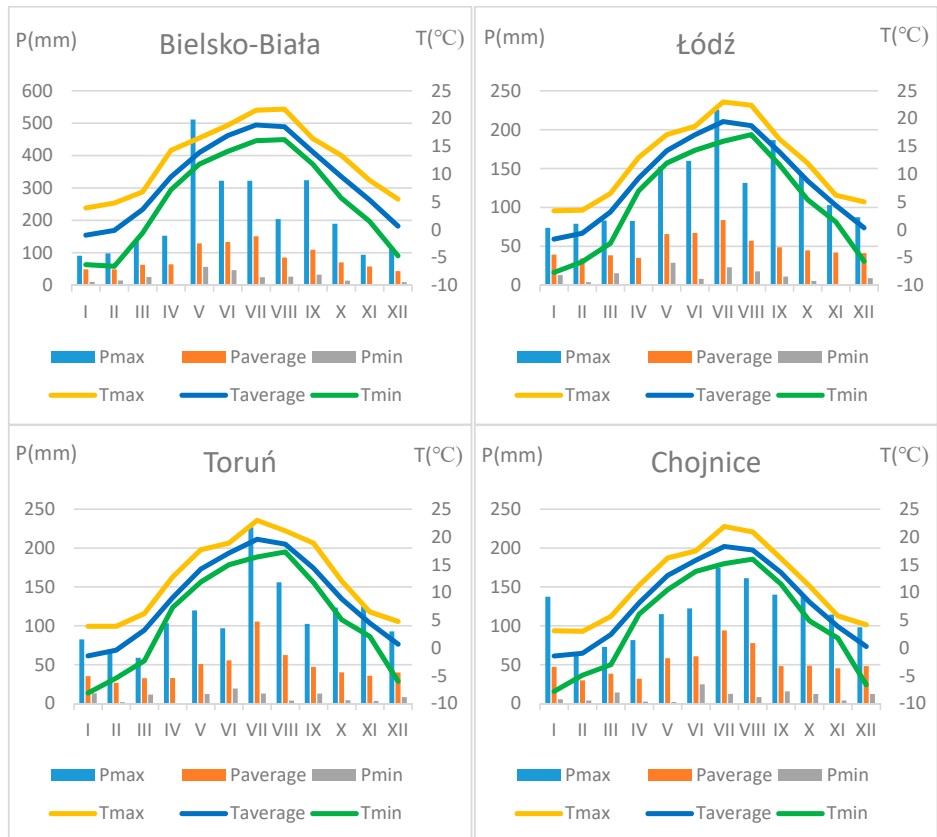

**Figure 3.** Average monthly precipitation (P) and average monthly air temperatures (T) in the period 1999–2018 (based on Institute of Meteorology and Water Management—National Research Institute data).

The highest average monthly precipitation occurred in July at three of the four analysed meteorological stations. The exception was the precipitation recorded at the meteorological station in Bielsko-Biała, where the highest monthly precipitation in the period 1999–2018 was recorded in May. The high average value was due to heavy rainfall in May 2010. The lowest average monthly precipitation during the year was recorded in January and February (Bielsko-Biała). Noteworthy are the months in which precipitation did not occur at all or did not exceed 1 mm. They took place in November (Bielsko-Biała, Łódź) and April (Toruń). The warmest months were July and August, in which the highest average monthly air temperatures were recorded, while the coldest were January and February.

*3.2. River Flows and Their Variability*

The size of Poland's water resources depends mainly on climatic conditions, which are variable. The measure of diversity of water resources is the specific runoff from the main Polish river basins. The average specific runoff in the years 1999–2018 was 5.36 $dm^3/s·km^2$ in the Vistula river basin, and 4.71 $dm^3/s·km^2$ in the Oder river basin. These values were comparable with the years 1951–2018. In dry years (2015 and 2018), the average specific runoff was up to 40% lower than the average specific runoff from the years 1999–2018 (Table 2). Compared to other rivers flowing into the Baltic Sea, Vistula and Oder are characterised by relatively low water resources [36]. The example of lowland rivers such as the Warta, Bug or Noteć shows that the average specific runoff from these catchments was even lower than that of the Vistula or the Oder, and amounted to 4 $dm^3/s·km^2$.

**Table 2.** Characteristics of the Vistula and the Oder runoff (based on Institute of Meteorology and Water Management—National Research Institute data).

| River | Hydrological Stations | Average Flow Rate in m³/s | | | | | Specific Runoff in dm³/s·km² | | | | |
|---|---|---|---|---|---|---|---|---|---|---|---|
| | | 1951–2018 | 1999–2018 | 2010 | 2015 | 2018 | 1951–2018 | 1999–2018 | 2010 | 2015 | 2018 |
| Vistula | Tczew | 1040 | 1029 | 1699 | 721 | 842 | 5.36 | 5.31 | 8.76 | 3.72 | 4.35 |
| Oder | Gozdowice | 517 | 487 | 795 | 297 | 371 | 4.71 | 4.43 | 7.24 | 2.71 | 3.38 |

The size of the specific runoff is varied in different regions of Poland. The highest specific runoff (12 dm³/s·km²) characterises the catchment areas of the mountain tributaries of the Vistula and Oder. The smallest specific runoff (4 dm³/s·km²) occurs in the catchment areas of the central lowlands, in the area of Kujawy and Wielkopolska, locally falling even to 1 dm³/s·km² [27]. The highest average annual specific runoffs were recorded in the Tatra Mountains (more than 50 dm³/s·km²), while in the Beskids they were between 15 and 20 dm³/s·km² [37]. The distribution of specific runoff in Poland depends on the sums of precipitation, as well as on underground supply and retention capacity of the catchment. The share of underground supply in Poland is on average 55%, i.e., in the runoff structure, it is more important than the surface supply. Significantly higher values of underground runoff are recorded in the northern part of Poland. Lake basins, similarly to the rivers of Przymorze, are characterised by a particular structure of runoff in which underground water supply prevails. On the other hand, in mountain areas, precipitation prevails. The retention and supply of groundwater has little influence on the structure of the runoff [38,39].

Figure 4 shows the size of flows of the Vistula and the Oder in the period 1999–2018. The largest flows occurred in 2010, and the lowest in 2015. The annual average values show a decreasing trend of flows in both cases in the analysed period. This is mainly due to an upward trend in air temperature and greater evaporation, despite a small upward trend in the sum of precipitation or lack of trend over the analysed multiannual period. The analysis of flow trends carried out over a longer period (65 years) showed a decreasing flow trend [40].

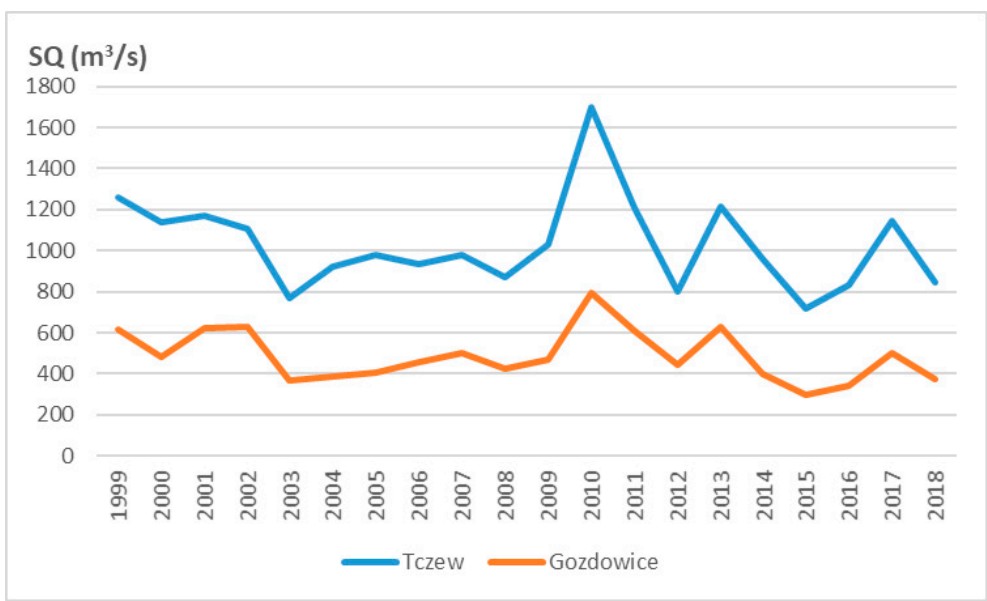

**Figure 4.** Average annual flows (SQ) of the Vistula in Tczew and the Oder in Gozdowice in the period 1999–2018 (based on Institute of Meteorology and Water Management—National Research Institute data).

Apart from annual variability, the monthly variability of flows plays a significant role. The lowest average monthly flows occurred in September and the highest in March and April (Figure 5). From the point of view of water resources use, low flow rates occur in the summer (from June to September),

when there is the highest air temperature, but also when there is the highest demand for water in the vegetation period. June was characterised by a high variability of flows, with flows up to several times higher or lower than the long-term average.

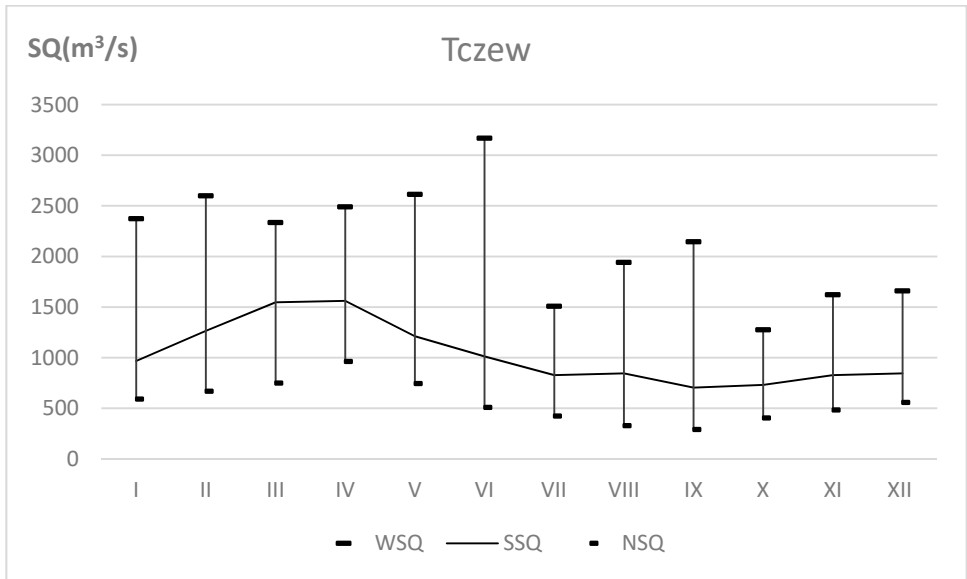

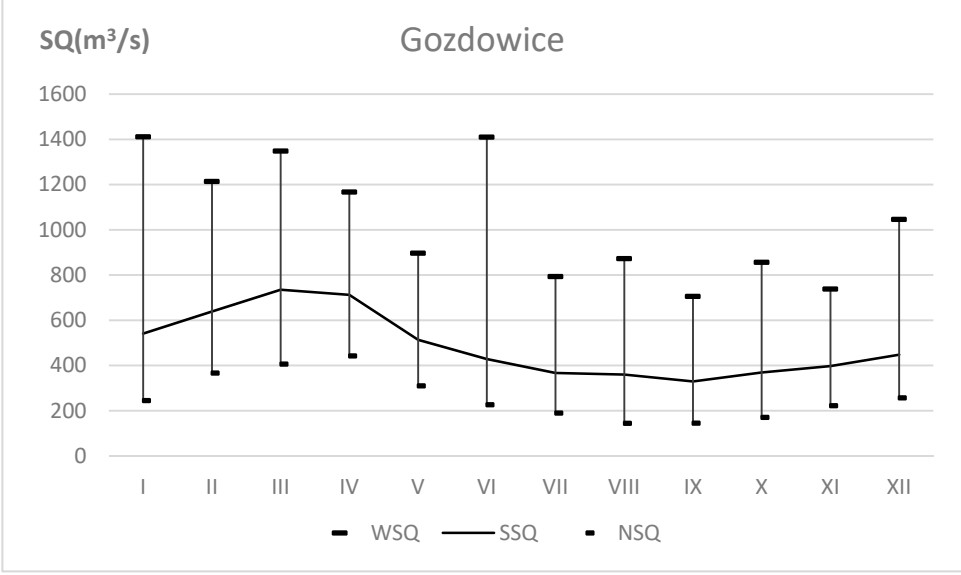

**Figure 5.** Maximum (WSQ), average (SSQ) and minimum (NSQ) monthly flows of the Vistula in Tczew and the Oder in Gozdowice in the period 1999–2018 (based on Institute of Meteorology and Water Management—National Research Institute data).

### 3.3. Water Needs of Poland

One of the most frequently used indicators of the amount of renewable water resources is the WSI (Water Stress Index). This indicator determines the amount of available resources per capita. A value of 1700 m$^3$/capita/year for renewable fresh water [16] has been proposed as a water scarcity threshold, below which social stress and high levels of competition for water appear. The resources of flowing water in Poland per capita during the year ranged from 1061 m$^3$/cap (2015) to 2255 m$^3$/cap (2010) (Figure 6). The average value in the years 1999–2018 was 1566 m$^3$/cap, which means that Poland is in the group of countries with low water resources, and exposed to water stress.

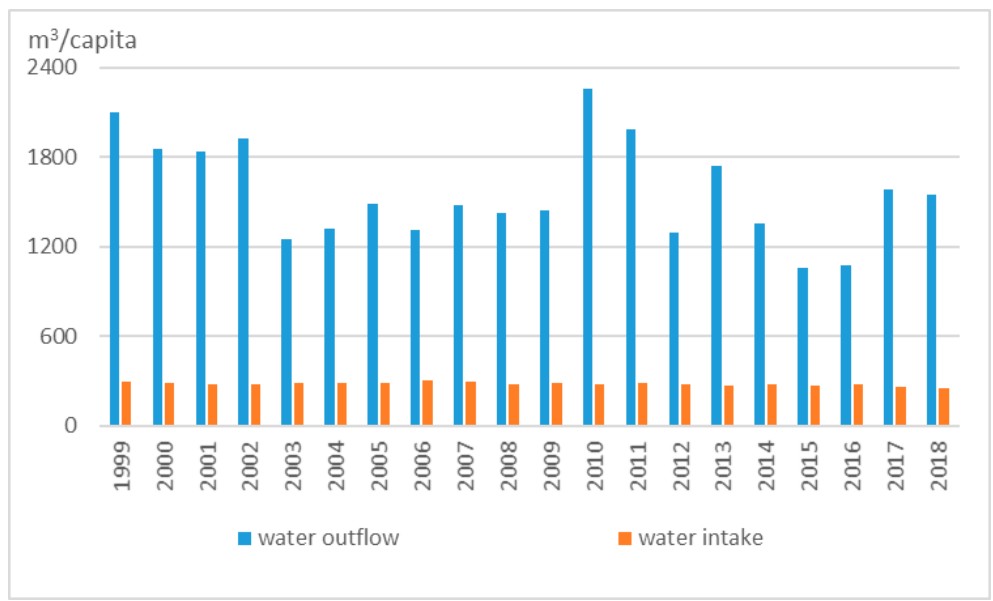

**Figure 6.** Volume of water outflow and intake in m³/cap in the period 1999–2018 (based on [26]).

The average water intake in the years 1999–2018 was 283 m$^3$ per capita. The highest intake was 310 m$^3$/cap and the lowest 257 m$^3$/cap. The volume of water intake per inhabitant over the last 20 years has remained stable. The variation in water intake changed by about 10% in relation to the average value from the years 1999–2018. This means that the volume of water consumption in relation to their resources has not been exceeded on an annual basis. However, it should be taken into account that part of the water used mainly for supplying the population with drinking water comes from groundwater intakes (due to its good properties), which are non-renewable or are undergoing slow recovery of water resources. In Poland, the average share of groundwater in the analysed period was 15.4%, surface water 83.8% and water from drainage 0.8% (Figure 7). The increase in groundwater intake is noticeable in dry periods, i.e., 2006, 2015, 2016. From 2014 to 2018, a systematic decrease in the amount of surface water intake compared to groundwater is recorded (Figure 8).

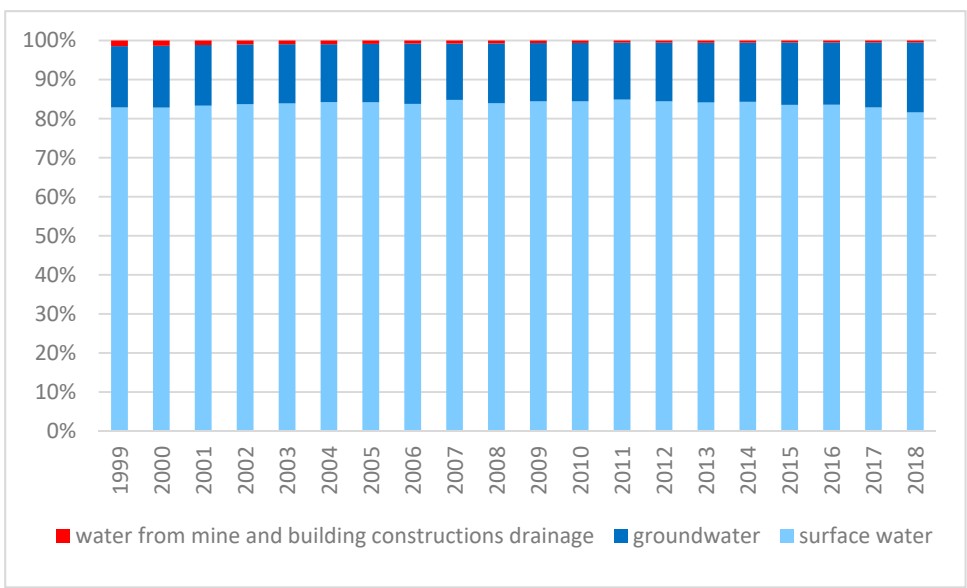

**Figure 7.** Water intake for the economy and population by source in the period 1999–2018 (based on [26]).

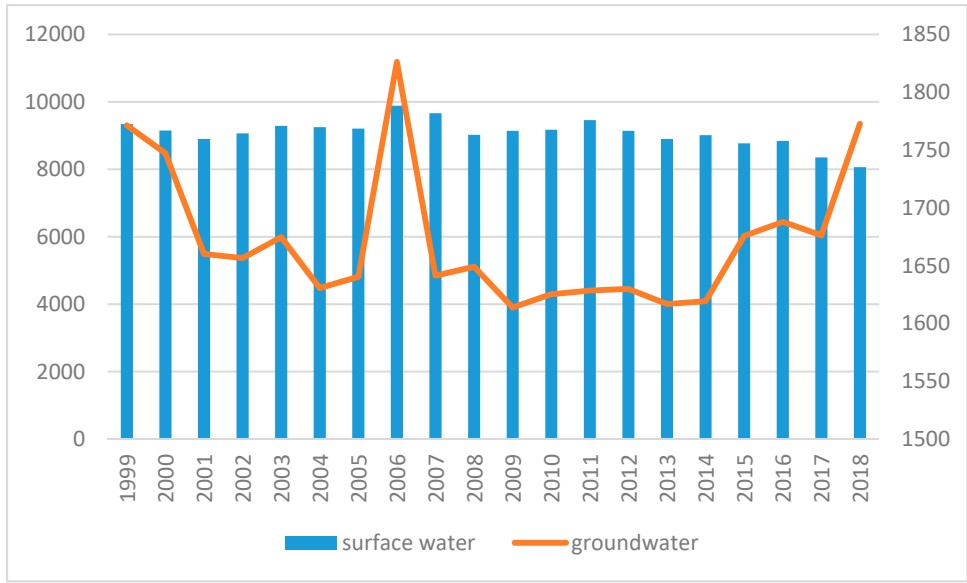

**Figure 8.** Water consumption (in km$^3$) divided into surface water and groundwater in the period 1999–2018 (based on [26]).

The Water Exploitation Index (WEI), which defines the ratio of water abstracted to total water resources, is another indicator commonly used to assess water scarcity (Figure 9). The advantage of this indicator is that it measures the amount of water used and relates it to available renewable water resources [41]. This indicator measures the pressure on water demand in the domestic, industrial and agricultural sectors in relation to local water supply. Areas with water scarcity can be identified on the basis of relative indicators of water demand exceeding 0.2, and the threshold of 0.4 (or 40% use relative to supply) signifying severely water-stressed conditions [42]. The combination of data on the water stress threshold allows the identification of water stress 'hot spots', areas where large numbers of people may suffer from water stress and its consequences.

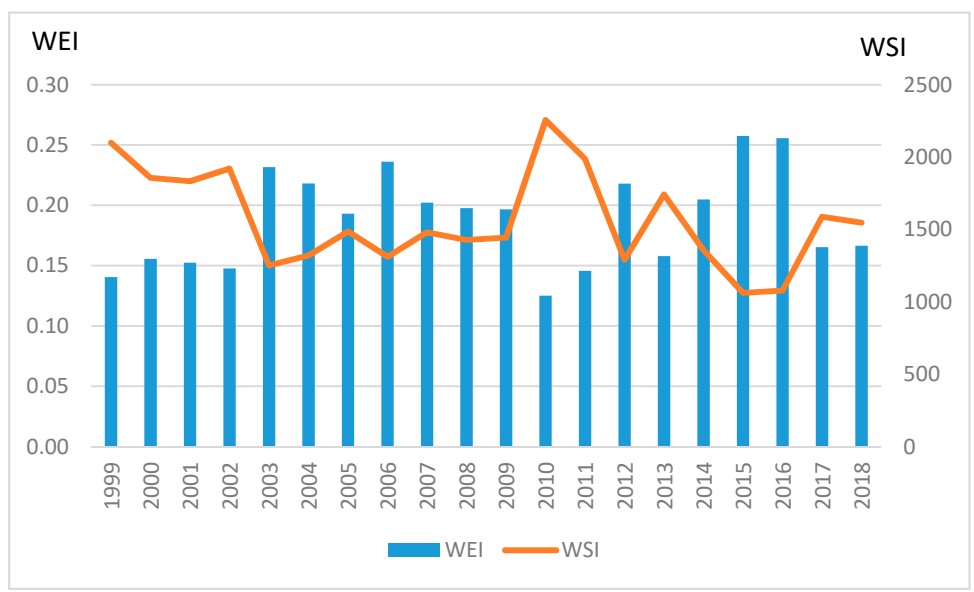

**Figure 9.** Water Stress Index (WSI) and Water Exploitation Index (WEI) in the period 1999–2018 (based on [26]).

The WEI indicator in the analysed period ranged from 0.13 (2010) to 0.26 (2015), which means that water consumption in relation to the average annual outflow from Poland was between 13 and 26%. In Poland this indicator is on average at the level of 0.19, which classifies Poland as an area with

moderate water stress. This results in restrictions on water use. In extremely dry years, the amount of available surface water will be significantly reduced. In such a situation, the water deficit ratio, which determines the ratio of water abstraction to available water, is in the range of 20–40%, classifying Poland as a country with medium to high water stress. Then, there is competition between water users, which may cause potential conflicts.

In the statistical statements of the Central Statistical Office, the structure of water intake takes into account the division into production purposes in industry, irrigation in agriculture and forestry, and filling and supplementing fish ponds, as well as exploitation of the water supply network (Figure 10). As a rule, production purposes include water intake for industrial purposes. Irrigation in agriculture and forestry and the filling of ponds include irrigation of areas of 20 ha and the filling of fishponds of at least 10 ha. This information is obtained on the basis of issued water consents and does not include irrigation for smaller areas. Similarly, no account is taken of quantities of water taken below 5 m$^3$/day, which are classified as normal water use and do not require water permits. The operation of the water supply network includes all the units that supervise the water supply network, including housing associations, water companies, water service companies, and production plants.

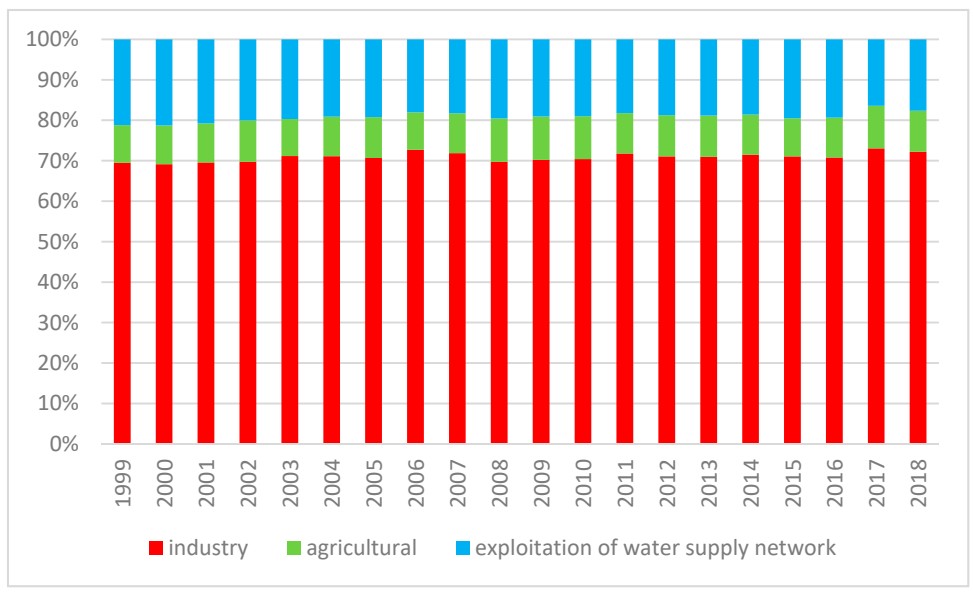

**Figure 10.** Structure of water consumption by the economy and population according to the sources of demand in Poland in the period 1999–2018 (based on [26]).

Industry had the largest share in water consumption, averaging 70.9% of total water consumption in the years 1999–2018. An average of 10.0% was allocated to irrigation in agriculture and forestry, while 19.1% went to the water supply network. During the analysed period of 1999–2018, total water consumption showed a significant downward trend. Water consumption for production purposes showed a slight downward trend, but it was not statistically significant. On the other hand, water consumption for the purposes of operating the water supply network showed a clear decrease. A slight increase in the amount of water intended for irrigation in Poland was noticeable, but compared to other European countries the amount was small. Agriculture in Europe uses around 40% of the total water consumed annually, while in the world it is around 70%. It is estimated that, despite increasing productivity, agriculture will continue to be the largest consumer in the coming years, thus increasing the water deficit in Europe. This is because more and more agricultural land needs to be irrigated, especially in Southern European countries [20,43,44]. Despite noticeable small downward and upward trends, the share of water consumption in Poland for particular sectors of the national economy was relatively stable and the recorded changes were small. Water consumption shows spatial differentiation and largely depends on the economic development of the region, which is already visible in the division

into voivodeships (Figure 11). The highest water consumption was recorded in the Mazowieckie, Świętokrzyskie, Zachodniopomorskie and Wielkopolskie Voivodeships. The reason for such high water consumption in these regions is the large water intake for industrial needs and the significant population of these regions. The lowest water intake was recorded in the Podlaskie and Lubuskie Voivodeships (less than 90,000 dam$^3$).

On the other hand, the largest amount of water intake for irrigation purposes was recorded in the Dolnośląskie, Lublskie and Kujawsko-Pomorskie Voivodeships. This was related to the predominance of agricultural activity over industrial activity in these areas [45]. During the three analysed years (2010, 2015 and 2018) the amount of water used was relatively stable (Figure 10). An increase in the amount of water used for irrigation purposes was recorded in the years with low precipitation (2015 and 2018) in the Kujawsko-Pomorskie Voivodeship, which has an exceptionally agricultural character. Voivodeships located in central Poland (Wielkopolskie, Mazowieckie, Kujawsko-Pomorskie) have very favourable soil conditions, which are the basis for intensive agricultural production.

It should be noted that these areas have the lowest precipitation during the year, and at the same time show significant water consumption for economic needs. This may lead to restrictions or destabilisation in the future socio-economic development of the region.

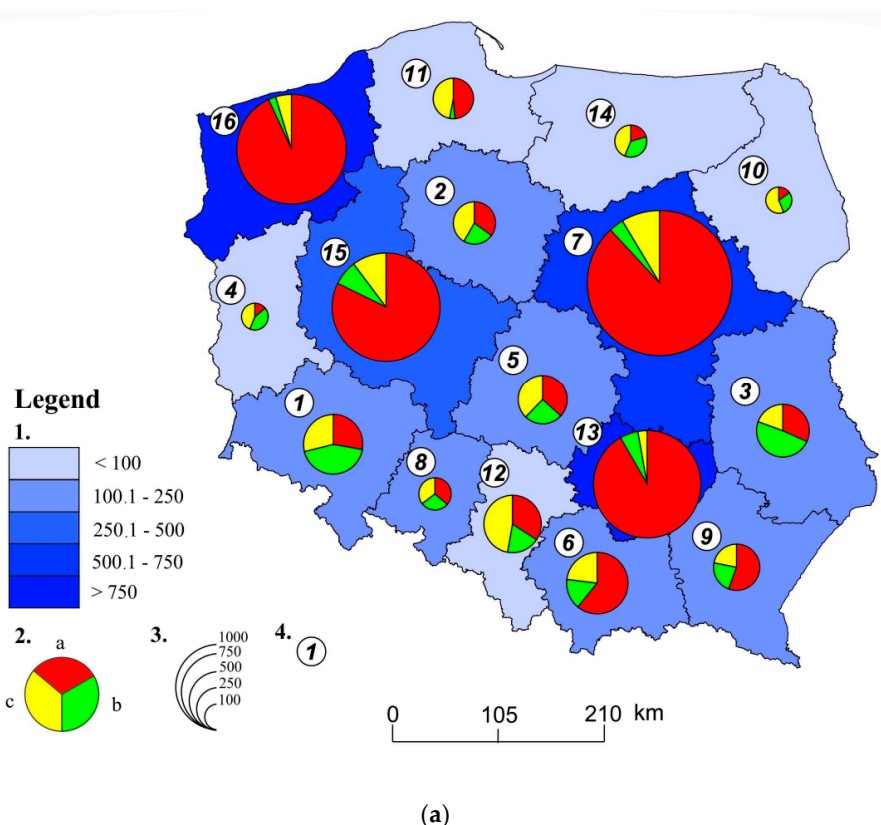

(**a**)

**Figure 11.** *Cont.*

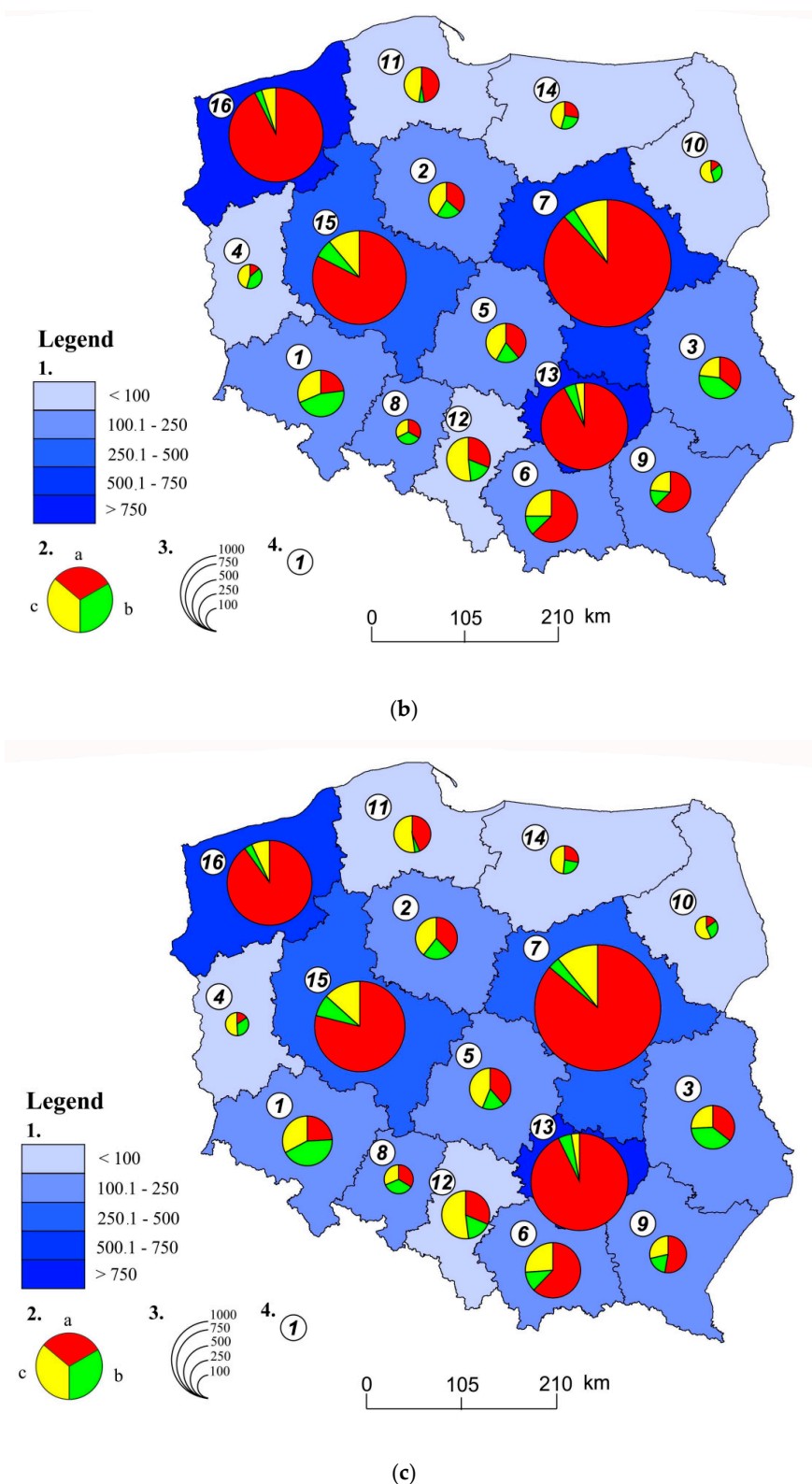

**Figure 11.** Structure of water consumption in Poland (by voivodeship) in 2010 (**a**), 2015 (**b**) and 2018 (**c**) (based on [26]). Explanations: 1—water consumption in m$^3$/capita; 2—share of % of water consumption for purposes: a—industry, b—operation of the water supply system, c—agriculture; 3—total water consumption in hm$^3$; 4—names of voivodeships according to the numbering (1—Dolnośląskie, 2—Kujawsko-Pomorskie, 3—Lubelskie, 4—Lubuskie, 5—Łódzkie, 6—Małopolskie, 7—Mazowieckie, 8—Opolskie, 9—Podkarpackie, 10—Podlaskie, 11—Pomorskie, 12—Śląskie, 13—Świętokrzyskie, 14—Warmińsko-Mazurskie, 15—Wielkopolskie, 16—Zachodniopomorskie).

## 4. Discussion

This paper analyses the variability of water scarcity caused by variability of meteorological conditions in Poland in 1999–2018. Meteorological, hydrological and statistical data were used to assess water resources and the degree of their use on an annual and monthly basis. The literature presents more than 150 different indicators that assess the degree of water use depending on the region of the world [42,46,47]. The study uses one of the most commonly used indicators, i.e., the Water Stress Index (WSI) proposed by Falkenmark and the Water Exploitation Index (WEI), interchangeably called the Water Deficit Index [16,18,41]. In the years 1999–2018, the average value of the WSI index for Poland was 1566 m³/cap/year, which qualified it as a country threatened by water deficit (water stress) (below 1700 m³/cap/year). On the other hand, the average value of the WEI index in the analysed period amounted to 0.19 (moderate water stress), which places Poland below the European average, i.e., it exceeds the critical level (0.16) considered safe by the European Environment Agency (EEA). In dry years, the WEI index amounted to 0.26, which qualifies Poland as a country with medium to high water stress, where there is competition between water users.

Poland's water resources are significantly spatially varied, which is a consequence of the changing natural environmental conditions shaping the hydrological cycle. Figure 12 is the water deficit map for 2015 for individual voivodeships. The calculations show that, taking into account the sum of precipitation and the lowest evaporation value (380 mm per year) and the amount of water abstraction, the voivodeships with the highest water deficits are Kujawsko-Pomorskie, Wielkopolskie, Mazowieckie and Opolskie. Evaporation exceeds precipitation over the year, which indicates a water shortage. Meanwhile, assuming evaporation of 550 mm during the year, only the Małopolskie Voivodeship did not have a water deficit, with renewable water resources exceeding evaporation and water consumption combined. In all 15 other voivodeships, water deficits were recorded, of which the highest were in Wielkopolskie, Mazowieckie, Kujawsko-Pomorskie and Łódzkie. These regions are the most exposed to water scarcity. This is particularly important as these are mainly agricultural regions, but also have developed industry and large populations, which means that the demand for water is high. In cases of surface water deficits, greater use of groundwater is observed.

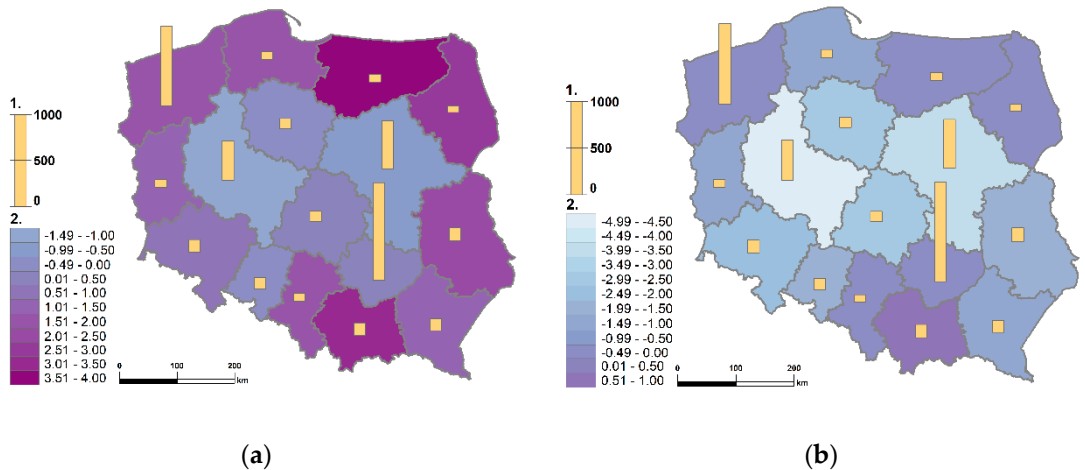

(**a**)   (**b**)

**Figure 12.** Water deficit map of Poland for 2015: (**a**) for 380 mm evaporation and (**b**) for 550 mm evaporation. Notes: Water consumption in m³/person, 2—water deficit in km³.

Progressing global warming is changing the water cycle and consequently the size of available water resources [48]. The results of studies on the impact of climate change on water resources in Poland so far indicate an increase in extremes of precipitation. Depending on the location of the meteorological station, precipitation showed different trends (increasing or decreasing) in the period 1951–2018, but in the shorter period 1999–2018 decreasing trends in annual precipitation sums prevailed. Comparative analysis of colder (1961–1990) and warmer (1991–2015) years showed that maximum

daily precipitation for the summer half-year increased at many meteorological stations in Poland and those increases are more numerous than those in the winter half-year [49]. The analysis carried out for selected meteorological stations revealed the occurrence of symptoms indicating systematic and persistent warming. The growth rate is different: from 0.18 °C to 0.34 °C per decade, which indicates that the average annual air temperature in Poland has increased by 1.1–2.2 °C since 1951. A significant increase is observed in average and extreme air temperatures. Studies [50,51] confirmed that the frequency of heat waves in Poland has increased, mainly in July and August (maximum temperature ≥ 25 °C).

These changes will adversely affect the size of water resources and their temporal and spatial distribution, limiting the availability of surface water and groundwater [52]. While the average annual outflow over longer research periods (1951–2018) remains unchanged or shows a slight decrease, in the case of seasonal changes it shows significant variability during the year. This is particularly true in the summer months, when river flows are generally low. The presented results for the period 1999–2018 are comparable to results obtained by other authors in multi-year periods [37,53–55]. The climate changes forecasts indicate that in the Oder and Vistula basins, the annual and seasonal runoff will increase significantly in the future. The seasonal patterns show the highest increase in winter and the lowest in spring, whereas the spatial patterns show the highest increase in the inner, lowland part, and the lowest in the southern, mountainous part of the basin [56]. Projected changes for Poland, estimated from the multi-model ensemble, showed that the annual temperature mean is expected to increase by approximately 1 °C during the period 2021–2050 and by about 2 °C during the period 2071–2100. This will not only have an impact on water resources but also on selected sectors such as the environment, agriculture and health [57].

The analysis of the variability of water resources in the Vistula and Oder river basins does not fully reflect the conditions in particular regions. The low variability of the outflow of the Vistula and Oder is a result of the delayed reaction of the catchment area to the increase in air temperature, evapotranspiration, precipitation and supply of underground waters. Long-lasting meteorological droughts that took place in 2015 led to a significant reduction or complete disappearance of flow in small watercourses in central Poland [58]. The reason was also the intake of water from rivers for the purposes of irrigation in agricultural areas. As a consequence of this type of treatment, additional water resources were extracted and underground water was drawn. As a result, a clear decrease in water resources was noticeable, and manifested in, among other things, reduced water retention in lakes, more frequent soil droughts and groundwater low-waters. The problem of surface water abuse and increasing dependence on non-renewable groundwater resources was recorded in various regions of the world. This raises serious doubts about sustainable water supply and food production in irrigated areas [18]. Reducing water abstraction and consumption is possible through the acquisition of effective technologies and adequate technical infrastructure related to water abstraction and wastewater disposal [59]. Therefore, many water managers and decision makers are actively pursuing management strategies on both the supply and the demand side [60].

Average annual values of water scarcity indicators do not fully reflect the variability and extent of water availability within the area during the year [2,61]. They completely disregard periods when water intake is impossible due to small quantities of water or inadequate water quality. During the drought of 2015, many areas introduced restrictions on the supply of water to industrial plants, reduced pressure in the water supply network, and a ban on filling garden swimming pools or watering lawns. Such warnings were issued for many regions of Poland. Water shortages were experienced locally by persons using individual shallow farm water intakes as well as in communes that used municipal water intakes using the first aquifer, and towns supplied with water from large surface water intakes. The rapid increase in demand for water with limited water resources requires rational water management at the national, regional and local levels [62]. Research is required on a local scale, at least on a monthly basis, to determine the scarcity of water resources taking into account the amount of water needed to maintain ecosystems.

A way to reduce the excess and deficit of water can be to undertake and implement small reservoir retention projects. Such works have been introduced in Poland by the small retention programme implemented by the State Forests in mountain and lowland areas in the years 2007–2015. Currently, three national programmes have been launched in Poland to support small retention in rural and urban areas (https://stopsuszy.pl/programy-wspomagajace-mala-retencje/#more-3533). The first programme, "Modernisation of farms—irrigation area on a farm", is a programme of subsidies for irrigation for farms. Farm owners can finance, among other things, the construction of wells and tanks and the purchase of machinery and equipment for the collection, storage, treatment, recovery or distribution of water, irrigation installations and irrigation control systems. The second programme is entitled 'City with Climate—Green-Blue Infrastructure'. It is a competition for local governments, which assumes co-financing of projects in the field of rainwater management and shaping urban greenery. There are no subsidies for private individuals under the programme. The third programme includes river-bed retention. It is a programme for shaping water resources in agricultural areas. The aim is to restore the double functionality of drainage facilities, which will ensure water retention on agricultural land during periods of drought. In many communes, there are local government programmes being implemented that subsidise pro-retentive activities.

A separate problem related to the use of water resources is the volume of water intake and consumption by different sectors of the national economy. Water consumption can be reduced in various ways. One of the simplest and at the same time most aggravating for all users is to increase water prices. Among individual water users, a popular form of water retention is the collection and use of rainwater in single-family houses. However, the most practical solution is to invest in the most modern technologies for water recovery and reuse in industrial plants, in agriculture and in newly constructed residential buildings. In Poland, unlike in other countries, there is little interest in building systems for the treatment and reuse of previously used water (so-called 'grey water'), which is mainly related to high installation costs. Grey water recycling means not only a significant reduction of operating costs but also measurable environmental benefits.

Reducing water consumption requires not only individual restrictions but also systemic water consumption solutions adapted to living conditions, and using the latest technology, in households. An effective tool is the proper management of water consumption, and water consumption at home should be seen in the context of social practices rather than individual behaviour [63].

## 5. Conclusions

Satisfying the growing demand for freshwater and at the same time protecting ecosystems with scarce and considerably variable water resources is one of the biggest challenges today. In this study, it was found that the average value of the so-called water resources in the Water Stress Index (WSI) in Poland in the years 1999–2018 amounted to 1566 $m^3$ per capita, which means that Poland is in the group of the countries with low water resources exposed to water stress. The average water consumption in 1999–2018 was 283 $m^3$ per capita and has remained relatively constant over the past 20 years. Taking into account water resources and the amount of water abstraction, the water Exploitation Index (WEI) was calculated, which in the analysed multiannual period was 0.19 (moderate water stress), placing Poland below the European average. Despite the small water resources of Poland among the remaining European countries, the amount of water resources is sufficient in terms of the country as a whole during the year. A significant problem is the seasonal variability of water resources over time and their spatial diversity in the country. They result from the high variability of precipitation and air temperature, which leads to the occurrence of extreme hydrological phenomena.

During the analysed 20 years (1999–2018), a decrease in the annual sums of atmospheric precipitations and an increase in average annual air temperatures were observed at selected meteorological stations in Poland. The rate of growth varies with the meteorological stations, ranging from 0.25 °C to the 0.40 °C per decade, indicating significant warming. Increasing air temperature causes higher evaporation, which affects the size of water resources. As a consequence,

a downward trend in the flows of the two largest rivers in Poland was observed. The intensity of hydrometeorological phenomena varies, as does the volume of water intake in individual regions of Poland. On the basis of the water deficit map, and taking into account the sums of precipitation, estimated values of evaporation (380 and 550 mm) and the volume of water abstraction for 2015, areas exposed to water deficit were identified. The calculations show that, taking into account the lowest evaporation value (380 mm during the year), the regions with the most water deficits include the Wielkopolskie, Mazowieckie, Opolskie and Kujawsko-Pomorskie voivodships. In other voivodships, the amount of available water was greater than the demand during the year. Assuming evaporation of 550 mm during the year, only the Małopolskie voivodship did not have a water deficit, and the amount of renewable water resources in this voivodship was higher than evaporation and water intake. Water deficit was recorded in the remaining 15 voivodships, of which the highest deficit was in Wielkopolskie, Mazowieckie, Kujawsko-Pomorskie and Łódzkie (central Poland). These four voivodships, covering central Poland, are characterized by low precipitation and high water intake for economic purposes.

In the light of the growing pressure to which water resources are being subjected, decisive measures should be taken to reduce the occurrence of water scarcity, especially in the summer period. In order to prevent overexploitation, integrated measures are required to reduce demand, minimise the amount of water abstracted and increase the efficiency of water consumption. Such measures should be taken primarily on a local and regional basis and should be tailored to the individual needs of individual municipalities and regions. However, this requires more detailed hydrometeorological monitoring and further research, which should primarily cover areas where water use is already large and water resources small.

**Author Contributions:** Conceptualisation, K.K.-W.; methodology, K.K.-W.; formal analysis, K.K.-W.; investigation, K.K.-W. and S.M.; writing—original draft preparation, K.K.-W.; writing—review and editing, K.K.-W. and S.M.; supervision, S.M. and K.K.-W.; project administration, K.K.-W.; funding acquisition, K.K.-W. All authors have read and agreed to the published version of the manuscript.

**Funding:** This research received no external funding

**Conflicts of Interest:** The authors declare no conflict of interest.

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
