# Peer review of "Influence of Climate Changes on the State of Water Resources in Poland and Their Usage"

_geosciences, doi:10.3390/geosciences10080312_

Round 1

Reviewer 1 Report

Influence of climate changes on the state of water resources in Poland and their usage

Short abstract ...
The authors made a statistical work about several climatic parameters, general water resources and water usage in Poland.
The Abstract sum up the main chapters followed in the paper in a synthetic/concentrated way.
The Introduction offers some ideas about water resources, in general, water demand and climate change with its challenges that affect the quality and quantity of available water for the population and ecosystems. The serious water management is absolutely necessary because of critical level touch by the balance between water demand and water availability. About the authors, Poland is one of the countries with the smallest water resources from Europe.
In the Material and Methods chapter - Research area, general informations - data about stable population of Poland, average population density, Poland relief, general climate and main Polish rivers were offered. Regarding the Data and Methods authors specified the type of statistical hydro and meteo used data and its origin, for the 1999-2018 period. Also, the main indicators regarding the Polish water resources and the way they are used, in general and regional perspective, were presented.
The Results chapter is, clearly, the most consistent. Here, the Variability in space and, especially, time of precipitation and temperature was presented and the main characteristics of these parameters at the used meteorological stations were offered. The most relevant graphs clearly indicated the trend of annual temperature and precipitation, and the monthly average graphs of these parameters shows their specificity for Polish territory. Concerning the variability of river flows, the authors concentrated on the main two collectors of Poland, Vistula and Oder. The average annual flows graph indicates an obviously negative trend of the parameter, that confirm the general tendency specified in the first part of study. Regarding the Water demand the same negative trend with annual oscillations can be distinguished on the specific graph along the 1999 and 2018 period, as the Water Stress Index (WSI) parameter. The Water Exploitation Index (WEI) presents an oscillating evolution, according the water demands in different sectors of economy and hidroedilitar. About the structure of water consumption by the economy and population there are present a relative stability of values along the studied period. Very expressive and useful is the spatial repartition of structure of water consumption, represented at national scale, where an evolution on different categories can be observed. The increase of water demand for any domains (eg. agriculture) validate the climatic trend that affect the Polish territory.
The Discussion chapter offers a supplementary interpretation on the analysed index and on the precipitation and water flows repartition in Poland under effect of global warming. Areas with water deficit and increasing water demand can be distinguished, and authors offers several solutions to reduce failure (eg. small reservoirs projects, rainwater management, water reusing, innovative technologies in households etc.).
Conclusions chapter closes the paper, offering a global image on the authors work, each chapter and subchapter being represented through the specific terms.

Detailed and specific observations ...
- lines 1-2 - because of relatively short period that you use for this study, the title is better to be modified in ”The state of water resources in Poland and their usage under the new climate evolution trends”; for identifying real climate changes elements and their correlation with liquid flow dynamics, the study period is recommend to be longest/extended;

- lines 51-52 - I don't know if Poland is the country with the smallest water resources from Europe!!!; if you say this, how do you think about Greece or other really poor in water countries?; with numerous lakes, rivers and a level of precipitation between 500 and 1100 mm, Poland seems not be the poorest country, regarding the water resources; probably, concerning the water quality, you have a little problem, because of eutrophication of waters etc., but quantitatively, I don't think; if you insist, you can say, ”one of the countries of Europe with quantitatively and qualitatively problems regarding the water resources”, but not more;

- lines 68-74 - the expression of type ”population density of 123 persons per 1 km2” is too much; the consecrated expression is ”population density of 123 persons per km2 or sq. km”; see the screen capture from Wikipedia, for example;

- line 78 - are you sure that in Poland mountainous area is starting from 500 m up to 2600 m?; 500 m seems to be, mostly, altitudes specific for hilly areas;

- line 105 - ”2 hydrometric or gauging stations”, instead of ”2 hydrologic stations”; hydrologic activity is more complex that hydrometric activity;

- lines 138-141 - the temperature map chromatic is not adequate for average temperature and for temperate areas; you can use another lowest intensity chromatic; an attachment is available;

- for all graphs with time on x axis - the presence of an equation of type y=0,0394x+8,7331 on a graph where one axis represents the time (years) is mathematically with problems; is better to remove just equation; time is not a parameter as the temperature, precipitation etc., to be correlated; you make only a distribution in time of any parameter, in hydrograph representation style; for example, please calculate the year (x value) depending on y value;

- for Fig. 6, you can add the two trends; this will help the reader;

- for a good compare possibility, all three figures (Fig. 10-12) is better to be on the same page; it is difficult to run from a page to another for making this;

- for animating and completing the Discussion chapter, is recommended to add one group of maps as attached type, that validate very well flow layer, necessary for different users layer and difference layer - practically, the overstock and poor areas; this will increase your paper quality;

Good work! Congrats.

Author Response

Thank you for your review. Responses to comments are provided in the Word file.

Reviewer 2 Report

  1. I suggest rewriting the abstract, which should include the brief contents of Research Objective, Methodology and Major Findings.
  2. As written in MS, the objective of this study was to estimate the amount of available renewable water resources of Poland in the period 1999-2018 and the degree of their use by various branches of the national economy in national and regional terms. HOWEVER, as the author said in the paper “The total amount of water resources was presented based on hydrological data from 2 hydrological stations. The volume of water intake by three sectors of the national economy (agriculture, industry and operation of the water supply network), total water consumption per capita and sources of water intake are presented. These values were obtained from the Local Data Bank of the Central Statistical Office (https://bdl.stat.gov.pl/BDL/start)”. It means the amount of available renewable water resources and the degree of their use by various branches were not originally estimated by this study.
  3. The new result might be the GIS-based maps to show the regional water consumption. If the study can show the same maps of regional available renewable water resources, then they can present the regional WSI and WEI. If so, this study will be perfect.
  4. The study pointed out: “there was no clear increase or decrease in annual precipitation” and “During the analyzed period of 1999-2018, total water consumption showed a significant downward trend”, however, “based on average annual flows of the largest rivers in Poland: the Vistula and the Oder. There was a clear downward trend in the flows”, why?

Author Response

(The authors gave the same response as above.)

Round 2

Reviewer 2 Report

I am so sorry to say that although the authors revised the MS by adding maps showing the deficit of surface water in a dry year, they still remain several doubts for me to make decision to accept this MS.

  1. The title has been changed to "The state of water resources in Poland and their usage under the new climate evolution trends", which means they should give some scenarios under the future climate change projections. Please make reference to "Assessment of climate change and associated impact on selected sectors in Poland" ( https://link.springer.com/article/10.1007/s11600-018-0220-4 ), which already gave the Model-supported projections of river runoff etc.

  2. This research just use constant values of evaporation, 380mm and 550mm to draw the maps of water deficit. However, as I know, there are many detail and high-resolution researches on the evaporation (ET) in Poland such as: "Estimation of Reference Evapotranspiration using the FAO Penman-Monteith Method for Climatic Conditions of Poland" (DOI: 10.13140/2.1.5080.4802), and  "Projections of changes of areal evapotranspiration for different land-use units in the Wielkopolska Region (Poland), https://link.springer.com/article/10.1007/s00704-016-1880-0 ). These researches showed that ET in Poland has regional differences as same as precipitation (P). If the authors just use a right map of P, but a false map of ET, I don't think their can output a right map of water deficit.

  3. I suggest that Figure 9 should give the amount of water consumption (WC) rather than the structure of WC. Then we can understand its annual changes.

  4. I suggest to use the same ranges in legends of Figure 11a and 11b, then we can understand the large differences between two scenarios.

  5. In abstract, the authors pointed out "During periods of surface water scarcity, groundwater uptake increases", however, this research did not give enough evidences to support this conclusion. 

  6. Please revise the conclusions to highlight the major findings of this study.

Author Response

Responses to the Reviewer's comments are below:

I am so sorry to say that although the authors revised the MS by adding maps showing the deficit of surface water in a dry year, they still remain several doubts for me to make decision to accept this MS.

  1. The title has been changed to "The state of water resources in Poland and their usage under the new climate evolution trends", which means they should give some scenarios under the future climate change projections. Please make reference to "Assessment of climate change and associated impact on selected sectors in Poland" (https://link.springer.com/article/10.1007/s11600-018-0220-4 ), which already gave the Model-supported projections of river runoff etc.
  • Change of the title to “The state of water resources in Poland and their usage under the new climate evolution trends” was proposed by Reviewer 1. We believe that the previous title "Influence of climate changes on the state of water resources in Poland and their usage" best reflects the nature of our study, which is why we returned to the original title of the work. In our study, we do not deal with future forecasts of climate change. In the study, we added the proposed literature on the projection of climate change and the impact on water resources.

  1. This research just use constant values of evaporation, 380mm and 550mm to draw the maps of water deficit. However, as I know, there are many detail and high-resolution researches on the evaporation (ET) in Poland such as: "Estimation of Reference Evapotranspiration using the FAO Penman-Monteith Method for Climatic Conditions of Poland" (DOI: 10.13140/2.1.5080.4802), and  "Projections of changes of areal evapotranspiration for different land-use units in the Wielkopolska Region (Poland), https://link.springer.com/article/10.1007/s00704-016-1880-0 ). These researches showed that ET in Poland has regional differences as same as precipitation (P). If the authors just use a right map of P, but a false map of ET, I don't think their can output a right map of water deficit.
  • The aim of our study was to estimate the amount of available renewable water resources and the article to indicate areas with water scarcity. We have not dealt with calculating the evaporation values ​​in different regions. Measurement of the amount of evaporation is carried out at a few meteorological stations in Poland, hence empirical formulas are used to determine its size, which use standard meteorological data (including air temperature, precipitation, solar radiation and others). More than 40 formulas for determining the amount of evaporation can be found in the literature. The obtained values ​​of evapotranspiration also depend on the adopted research period, research area, land use and other factors.

The study assumed 2 evapotranspiration values ​​of 380 mm (lower) and 550 mm (higher). These values ​​were adopted in accordance with the study by J. Kołodziej (2008), who indicated that the annual sum of evaporation in Poland was usually in the range from 380 to 550 mm. The adopted evaporation values ​​are similar to the results obtained by other authors. In the Hydrological Atlas of Poland (1987), evaporation calculated by the Konstantinov method was from 440 to 540 mm in Poland, with the predominance of evaporation in most of the area from 480 to 560 mm. Jokiel (2007) reported for the area of ​​central Poland that the evaporation was from 480 to 520 mm. According to Jankowiak and Kędziora (2009), the average evaporation in Wielkopolska was 525 mm during the year. More detailed studies were carried out by Łabędzki et al. 2011 for the entire territory of Poland during the growing season (April-September). The detailed amount of evaporation taking into account land use for the area of ​​Wielkopolska was presented by Szwed (2017), according to which the average amount of evaporation per year was 583 mm. These items were added to the reference list.

In most studies, the differences in the amount of evaporation between different regions of Poland were relatively small. Łabędzki et al. (2011) calculated the amount of evaporation based on the FAO Penman-Monteith method, on average for Poland from 35 years (1970-2004) and 40 stations, it was 520 mm in the growing season from April to September (6 months a year). According to Łabędzki et al. (2011), the spatial variability of reference evapotranspiration was lower than the temporal variability. It means the bigger differentation of the reference evapotranspiration among years than among stations.

The adopted evaporation values ​​of 380 and 550 mm are in line with the tests carried out by others Considering the above, it should be considered that the evaporation values ​​adopted for the estimation are similar to the results of other authors and do not differ significantly from the values ​​of so far recognized. The water deficit map shown includes estimates of water availability but is not a map that represents false evaporation values.

Atlas Hydrologiczny Polski, Wydawnictwo Geologiczne, Warszawa, 1987

Jankowiak J, Kędziora A. Globalne zmiany klimatu i ich wpływ na rolnictwo w Polsce (Global climate changes and their influence on agriculture). In: Zegara JS (ed) Z badań nad rolnictwem społecznie zrównoważonym (From the research on socially balanced agriculture). IERiGZ-PIB, Warszawa, p 101, 2009

Jokiel P. Zmiany, zmienność i ekstremalne sumy parowania terenowego i ewapotransporacji potencjalnej w łodzi w drugiej połowie XX wieku (Changes, changeability  and extreme total Surface evaporation and potential evaprotanspirations in Łódź in 2nd half on the XX th century). Acta Universitatis Lodziensis 2007, 63-88

Łabędzki, L.; Kanecka-Geszke, E.; Bąk, B.; Słowińska, S. Estimating reference evapotranspiration using the FAO Penman-Monteith method for the climatic conditions on Poland. In: Łabędzki L. (ed.) Evapotranspiration, Rijeka, InTech, 2011, pp. 275–294.

Szwed, M. Projections of changes of areal evapotranspiration for different land-use units in the Wielkopolska Region (Poland). Theor Appl Climatol 2017, 130, 291–304. doi:10.1007/s00704-016-1880-0

  1. I suggest that Figure 9 should give the amount of water consumption (WC) rather than the structure of WC. Then we can understand its annual changes.
  • We have inserted an additional graph (Fig. 8) which shows water consumption broken down by the amount of surface and groundwater.
  1. I suggest to use the same ranges in legends of Figure 11a and 11b, then we can understand the large differences between two scenarios.
  • We have improved the legend in the maps

  1. In abstract, the authors pointed out "During periods of surface water scarcity, groundwater uptake increases", however, this research did not give enough evidences to support this conclusion. 
  • This conclusion was formulated on the basis of Fig. 7 and Fig. 8. In particular, Fig. 8 shows an increase in the amount of groundwater abstracted in dry years, e.g. 2006, 2015 or 2018.
  1. Please revise the conclusions to highlight the major findings of this study.
  • We improved the conclusions.

Round 3

Reviewer 2 Report

The MS has been revised several times according to my reviewing comments, and it can be reconsidered for publication after minor revision.

  1. Please remove all regression equations in the all figures, because all these equations have no statistical meaning. 

  2. Characters in Figure 12 are too small to be read. Please use same data range in the legend of Figure 12.

  3. Please do a native English Check to the MS before publication.

Author Response

The MS has been revised several times according to my reviewing comments, and it can be reconsidered for publication after minor revision.

  1. Please remove all regression equations in the all figures, because all these equations have no statistical meaning. 
  • We deleted the equations and we improved the graphs.
  1. Characters in Figure 12 are too small to be read. Please use same data range in the legend of Figure 12.
  • We corrected the legend
  1. Please do a native English Check to the MS before publication.
  • The paper has been checked by a native speaker

Thank you for your comments.
